# Bevacizumab-Containing Chemoimmunotherapy for Recurrent Non-Small-Cell Lung Cancer after Chemoradiotherapy: Case Report

**DOI:** 10.3390/medicina57060547

**Published:** 2021-05-29

**Authors:** Nobutaka Kataoka, Yusuke Kunimatsu, Rei Tsutsumi, Nozomi Tani, Izumi Sato, Mai Tanimura, Takayuki Nakano, Keiko Tanimura, Daishiro Kato, Takayuki Takeda

**Affiliations:** 1Department of Respiratory Medicine, Japanese Red Cross Kyoto Daini Hospital, Kyoto 602-8031, Japan; nkataoka@koto.kpu-m.ac.jp (N.K.); ky92020223@yahoo.co.jp (Y.K.); r-sai@koto.kpu-m.ac.jp (R.T.); nozomi-t@koto.kpu-m.ac.jp (N.T.); izumi-h@koto.kpu-m.ac.jp (I.S.); mai.tanimura@outlook.jp (M.T.); tnakano@koto.kpu-m.ac.jp (T.N.); keiko-t@koto.kpu-m.ac.jp (K.T.); 2Department of Thoracic Surgery, Japanese Red Cross Kyoto Daini Hospital, Kyoto 602-8031, Japan; dkatoh@koto.kpu-m.ac.jp

**Keywords:** bevacizumab, chemoimmunotherapy, chemoradiotherapy, non-small-cell lung cancer, recurrence

## Abstract

Chemoimmunotherapy has become the standard of care as the first-line treatment of advanced or recurrent non-small-cell lung cancer (NSCLC). The bevacizumab-containing chemoimmunotherapy regimen is theoretically more effective than a non-bevacizumab-containing regimen via two mechanisms: a superior outcome of bevacizumab-containing chemothrerapy than the standard platinum doublet regimen, and the synergistic effect of bevacizumab with an immune checkpoint inhibitor (ICI). Bevacizumab effectively normalizes vascularization, especially when the vascular bed is damaged by previous treatment. Bevacizumab promotes immunomodulation when used with ICI. We describe a patient with nonsquamous NSCLC who returned 2.5 years after definitive chemoradiotherapy for postoperative locoregional recurrence in the right supraclavicular lymph node. Considering the destroyed vascular bed due to prior chemoradiotherapy, attaining vascular normalization was critical for effective drug delivery. The patient was treated with a bevacizumab-containing chemoimmunotherapy regimen, which resulted in a complete metabolic response. The patient responded well for 23 months and is receiving ongoing treatment. Thus, bevacizumab-containing chemoimmunotherapy could be advantageous in some recurrent cases after chemoradiotherapy.

## 1. Introduction

Chemoimmunotherapy combining cytotoxic chemotherapy and immune checkpoint inhibitors (ICIs) has been established as the standard of care as the first-line treatment of advanced or recurrent non-small-cell lung cancer (NSCLC). It addresses the high progressive disease (PD) rate of ICI monotherapy [1,2,3,4]. The addition of bevacizumab to standard chemotherapy with carboplatin and paclitaxel (bevacizumab plus CP; BCP) was more effective compared to CP monotherapy in terms of increasing overall survival (OS), median progression-free survival (PFS), and response rate in patients with nonsquamous NSCLC [5]. The addition of atezolizumab to BCP (ABCP) as the first-line treatment significantly improved PFS and OS [3]. However, it is unclear whether the ABCP regimen applies to heavily treated cases, especially recurrent cases after definitive chemoradiotherapy.

While the efficacy of ICI monotherapy on recurrent cases after chemoradiotherapy has been already reported, there is no publication regarding the efficacy of chemoimmunotherapy under the same condition.

This study reports a patient with nonsquamous NSCLC, who returned 2.5 years after concurrent chemoradiotherapy (CCRT) for a postoperative locoregional recurrence in the right supraclavicular lymph node, treated with the ABCP regimen, which resulted in a complete metabolic response (CMR).

## 2. Case Report

A 69-year-old male patient presented with adenocarcinoma without an oncogenic driver mutation or fusion, including *EGFR*, *ALK*, *ROS*1, and *BRAF*. The adenocarcinoma was associated with programmed death-ligand 1 (PD-L1) expression of 90–100%. The patient had undergone right upper lobectomy (pT2bN2M0, stage IIIA). This had been followed by adjuvant chemotherapy with a combination of cisplatin, pemetrexed, and bevacizumab for four cycles. Locoregional recurrence was documented in the right supraclavicular lymph node three years after surgery, and had been treated with definitive CCRT (60 Gy in 30 fractions with two cycles of carboplatin plus tegafur/gimeracil/oteracil potassim [S-1]), resulting in a complete response. 

However, 2.5 years after definitive CCRT, ^18^F-fluorodeoxyglucose (FDG) positron emission tomography (PET)/computed tomography (CT) revealed the same right supraclavicular lymph node swelling with a maximum standardized uptake value (SUV_max_) of FDG scoring 3.3 (Figure 1A). He was then referred to our department for further treatment.

Upon reevaluating the patient’s status, the right supraclavicular lymph node measured 7.4 mm in the short diameter on contrast-enhanced CT (Figure 1B) without any distant metastasis. The multidisciplinary lung cancer tumor board concluded that this was the second locoregional recurrence after definitive CCRT for the first postoperative recurrence, for which any curative treatment was not an option. He received four cycles of an induction ABCP regimen, resulting in partial response (PR) after two cycles (Figure 2A). This caused a metabolic PR with the SUV_max_ of FDG decreasing to 1.5 after four cycles (Figure 2B). Continuation maintenance therapy (CMT) with a combination of atezolizumab and bevacizumab was administered, and PET/CT after 11 cycles of CMT demonstrated CMR without FDG accumulation (Figure 2C). The patient responded well to the treatment for 23 months after initiating the ABPC regimen and is receiving 25 cycles of CMT. The patient experienced no severe adverse events during the observational period.

## 3. Discussion

ICIs have changed the treatment strategy for NSCLC, and pembrolizumab monotherapy has become the standard first-line treatment for advanced or recurrent NSCLC with PD-L1 expression ≥50% [6]. Therefore, pembrolizumab monotherapy, which is applicable to previously treated cases, could have been one treatment option in the current case with high PD-L1 expression. However, a high PD rate (>20% regardless of race) was detected in this population, which poses an urgent problem. High rates of PD reaching 28.7% among French patients [7] and 24.2% among Japanese patients [8] have led to early PD in the population. The early PD rate was reported to be similar to that of populations with ultrahigh PD-L1 expression of 90 to 100% [9]. This finding contradicts another report [10]. Chemoimmunotherapy theoretically addresses the high PD rate of ICI monotherapy by converting immune-desert or immune-excluded tumors into immune-inflamed status [11]. Thus, chemoimmunotherapy was considered in the current case, though the patient had received adjuvant chemotherapy as well as CCRT after the first locoregional recurrence. Among several chemoimmunotherapy regimens, the ABPC regimen was finally adopted considering the benefit of bevacizumab addition in the current case, which is discussed below.

Meanwhile, bevacizumab is more effective when combined with platinum-based chemotherapy through antiangiogenesis and vascular normalization [12]. Although the E1505 trial showed that the addition of bevacizumab to adjuvant chemotherapy did not improve OS after curative resection [13], the current patient received bevacizumab containing adjuvant chemotherapy before the final result of the E1505 trial. Vascular normalization is vital for improving intratumoral blood flow, drug delivery, and responsiveness to chemotherapy. Furthermore, bevacizumab also improves treatment efficacy by promoting the immune-modulating effects of dendritic cell maturation and T cell response activation, which are the key elements in the priming phase. They also accelerate the effector phase through trafficking and infiltration of T cells into the tumor, and reducing myeloid-derived suppressor cells (MDSCs) and regulatory T cells (Tregs) [11,14,15]. Since MDSCs and Tregs play an essential role in immune tolerance, bevacizumab enhances the effect of atezolizumab in the immune-inflamed status by restoring cancer immunity.

The superiority of the ABCP regimen over the BCP regimen has been confirmed in a phase III trial [3], even in situations wherein the reference arm of BCP was considered more effective than the conventional platinum doublet chemotherapy [5]. 

Regarding the recurrent cases after definitive chemoradiotherapy, a retrospective study of anti-PD-1 therapy demonstrated that the objective response ratio and median PFS of ICI monotherapy were 45.0% and 8.4 months, respectively [16]. This study shows the limited efficacy of ICI monotherapy in recurrent cases after chemoradiotherapy. The high PD rate of ICI monotherapy for previously treated NSCLC could be one major reason [17,18,19]. This could also be theoretically explained by the damaged vascular bed of the tumor after radiotherapy, which leads to the immune-excluded status with impaired ICI delivery and CD8 positive T cells in the target tumor [11]. Thus, deploying bevacizumab in combination with ICI in recurrent cases after CCRT could be advantageous.

Since there are no head-to-head trials comparing chemoimmunotherapy regimens as the first-line treatment, there are no data comparing each chemoimmunotherapy regimen. 

In the current heavily treated case, an ABCP regimen consisting of bevacizumab was adopted, considering the devastated vascular bed due to prior chemoradiotherapy, though it remains a matter of speculation. However, CMR was documented after 11 cycles of atezolizumab plus bevacizumab–CMT, which suggests that the aforementioned vascular normalization and immunomodulation by bevacizumab might have caused the synergistic effect.

## 4. Conclusions

The ABCP regimen is usually applied as the first-line treatment. However, it was helpful in previous cases with specific situations, consistent with the theoretical basis delivered in detail in the Discussion. The PFS in the current case has reached 23 months with an ongoing regimen, which surpassed the 8.4 months of median PFS of the ICI monotherapy for the recurrent cases after definitive chemoradiotherapy [16]. The current case suggested the utility of the ABCP regimen among suitable patients who experienced recurrence after CCRT. The application of chemoimmunotherapy can become more flexible based on the underlying mechanisms.

## Figures and Tables

**Figure 1 medicina-57-00547-f001:**
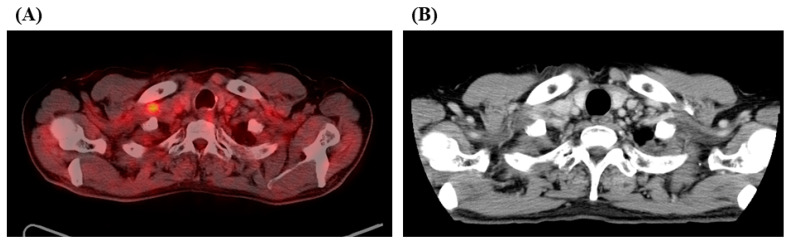
^18^F-fluorodeoxyglucose (FDG) positron emission tomography/computed tomography (CT) obtained 2.5 years after definitive chemoradiotherapy (**A**) demonstrated a right supraclavicular lymph node swelling with a maximum standardized uptake value of FDG scoring 3.3. Contrast-enhanced CT (**B**) showed a right supraclavicular lymph node swelling of 7.4 mm in shorter diameter.

**Figure 2 medicina-57-00547-f002:**
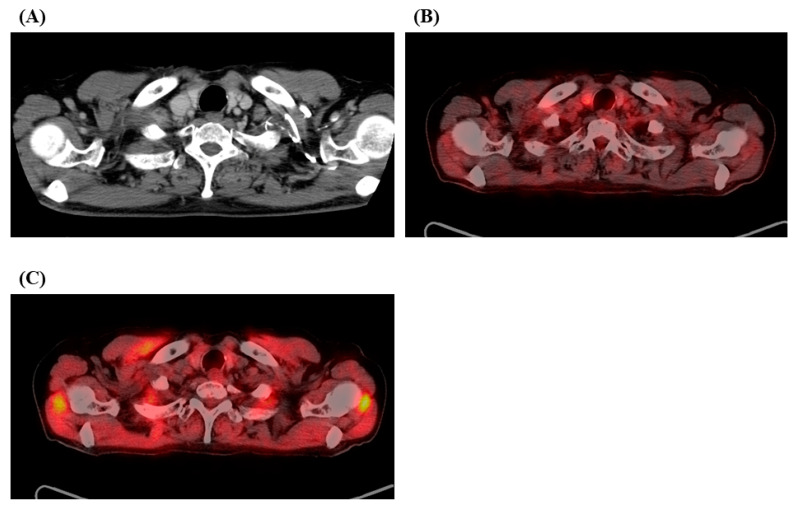
Contrast-enhanced CT after two cycles of induction therapy (**A**) with atezolizumab, bevacizumab, carboplatin, and paclitaxel (ABCP) demonstrated partial response (PR), and ^18^F-fluorodeoxyglucose (FDG) positron emission tomography/CT after four cycles of induction therapy (**B**) showed metabolically PR with a maximum standardized uptake value of FDG decreasing to 1.5. FDG/PET after 11 cycles of continuation maintenance therapy (**C**) with atezolizumab and bevacizumab exhibited complete metabolic response without any FDG accumulation.

## Data Availability

The data supporting the conclusion are included in the article.

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
