# Peer review of "Bevacizumab-Containing Chemoimmunotherapy for Recurrent Non-Small-Cell Lung Cancer after Chemoradiotherapy: Case Report"

_medicina, 2021, doi:10.3390/medicina57060547_

Round 1
Reviewer 1 Report
This is an interesting case report. However, there are some controversies. For example, the use of bevacizumab in the adjuvant setting and the disease extension, too. In particular, the patient presented an oligoprogression. The use of a loco-regional treatment may be appropriate in this context. Moreover, a single lymph node measuring 7.4 mm in diameter on contrast-enhanced CT with a maximum standardized uptake value of
FDG scoring 3.3 is uncertain disease progression. Please, explain the aspects in the discussion this is an interesting case report. However, there are some controversies. For example, the use of bevacizumab in the adjuvant setting and the disease extension, too. In particular, the patient presented an oligoprogression. The use of a loco-regional treatment may be appropriate in this context. Moreover, a single lymph node measuring 7.4 mm in diameter on contrast-enhanced CT with a maximum standardized uptake value of
FDG scoring 3.3 is uncertain disease progression. Please, explain the aspects in the discussion this is an interesting case report. However, there are some controversies. For example, the use of bevacizumab in the adjuvant setting and the disease extension, too. In particular, the patient presented an oligoprogression. The use of a loco-regional treatment may be appropriate in this context. Moreover, a single lymph node measuring 7.4 mm in diameter on contrast-enhanced CT with a maximum standardized uptake value of
FDG scoring 3.3 is uncertain disease progression. Please, explain the aspects in the discussion.
Author Response
Response:
I really appreciate your sincere suggestions and advices, and I agree with your concerns.
As you mentioned, bevacizumab containing chemotherapy is not standard in the adjuvant setting. This regimen was adopted by the surgeons and I cannot know the reason.
Thereafter, the locoregional recurrence in the right supraclavicular lymph node was documented, and the patient received definitive chemoradiotherapy. Two and a half years after the definitive chemoradiotherapy, the relevant recurrence in the right supraclavicular lymph node was observed.
As you mentioned, a single lymph node swelling measuring 7.4mm in diameter could be regarded as non-specific. However, this was the short diameter and the long diameter was 11.5mm. Considering that non-specific lymph node swelling is uncommon in the supraclavicular lymph nodes compared to other lymph nodes such as inguinal lymph nodes, a multidisciplinary tumor board at our hospital concluded that this is the second locoregional recurrence.
I added some explanation in order to clarify your concerns, avoiding redundancy.
Reviewer 2 Report
The article “Bevacizumab containing chemoimmunotherapy for recurrent non-small-cell lung cancer after chemoradiotherapy” presents a case report where the ABCP regimen was used in a pre-treated lung cancer patient.
- The fact that the paper is a case report should be clearly stated in the title. The CARE checklist should be included.
- Although all the important landmarks of the case are mentioned, the timeline is somewhat fuzzy and can be imrpoved.
- Why did the patient receive Bevacizumab as an adjuvant treatment? To my knowledge, this has never been standard. Also, why did he not receive maintenance Bevacizumab (as in the E1505 trial) after the four cycles of chemotherapy? Also, after the first relapse, why did the patient receive only two cycles of treatment? How did the patient tolerate the treatment?
- In the Discussions section, the phrases “However, a high PD rate (>20% regardless of race) was detected in this population, which poses an urgent problem. High rates of PD rates reaching 28.7% among French patients [7]and 24.2% among Japanese patients [8] have led to early PD in the population.” make no sense. Please rephrase. Also, the discussion section should offer more data regarding other cases in the literature and comment the importance of pre-treatment in the setting of ABCP regimen.
- I think the authors should highlight the uniqueness / the importance of their case in order to make the paper more interesting for the reader.
- Minor comments:
- Abstract - “The bevacizumab regimen is theoretically more effective via two mechanisms: a superior outcome than the standard platinum doublet regimen, confirmed in a phase III trial, and the synergistic effect of bevacizumab with an immune checkpoint inhibitor (ICI).” – I think the meaning of this phrase is difficult to comprehend, especially since the authors state that bevacizumab is more effective, but not compared with what. Please rephrase.
- Case Report – “The patient had undergone right upper lobectomy (pT2bN2M0, stage IIIA) six years preceding the start of this study.” This is not a study. Please rephrase and try to create a more clear timeline.
- Discussion – “ICI have changed the treatment” instead of “ICI has changed the treatment”
Author Response
I really appreciate your sincere suggestions and advices.
I have prepared the responses to your concerns in a point-by-point manner, which are attached below.
I am willing to respond as soon as possible if you have further concerns.
The article “Bevacizumab containing chemoimmunotherapy for recurrent non-small-cell lung cancer after chemoradiotherapy” presents a case report where the ABCP regimen was used in a pre-treated lung cancer patient.
1. The fact that the paper is a case report should be clearly stated in the title. The CARE checklist should be included.
Response:
I appreciate your advice and added “case report” in the title. I also attached the CARE checklist.
2. Although all the important landmarks of the case are mentioned, the timeline is somewhat fuzzy and can be imrpoved.
Response:
I am sorry for the unclear timeline, and reworded the Case Report section.
3. Why did the patient receive Bevacizumab as an adjuvant treatment? To my knowledge, this has never been standard. Also, why did he not receive maintenance Bevacizumab (as in the E1505 trial) after the four cycles of chemotherapy? Also, after the first relapse, why did the patient receive only two cycles of treatment? How did the patient tolerate the treatment?
Response:
I really appreciate your sincere suggestion.
As you mentioned, bevacizumab containing chemotherapy is not standard as an adjuvant treatment. This regimen was adopted by the surgeons and I cannot know the reason. The patient did not receive bevacizumab continuation maintenance therapy because this was added as an adjuvant treatment, in which surgeons considered the cure was obtained at the moment and bevacizumab maintenance therapy is not recommended according to the results from E1505 trial, as you pointed out.
Regarding two cycles of carboplatin plus S-1, this regimen was used as concurrent chemoradiotherapy and two cycles is standard. I agree with you in that this point is unclear, I reworded the relevant parts.
4. In the Discussions section, the phrases “However, a high PD rate (>20% regardless of race) was detected in this population, which poses an urgent problem. High rates of PD rates reaching 28.7% among French patients [7]and 24.2% among Japanese patients [8] have led to early PD in the population.” make no sense. Please rephrase. Also, the discussion section should offer more data regarding other cases in the literature and comment the importance of pre-treatment in the setting of ABCP regimen.
Response:
I appreciate your suggestion. We added the relevant phrases in order to make sure that ICI monotherapy for the patients with high PD-L1 expression cannot expect enough efficacy, which could be one therapeutic option in the current case with recurrent NSCLC with high PD-L1 expression. I agree with you in that the intent of these phrases on the discussion is unclear, and I added a sentence “Therefore, ICI monotherapy could have been one treatment option in the current case” before the relevant phrases to clarify the intent.
As to the pre-treatment in the setting of ABPC regimen, there is no report on the efficacy of ABPC regimen in cases with previously treated NSCLC. The confirmed efficacy of ABPC regimen remains the chemotherapy naïve NSCLC. Then, I added some sentences in the Discussion section to strengthen this point.
5. I think the authors should highlight the uniqueness / the importance of their case in order to make the paper more interesting for the reader.
Response:
I appreciate your sincere suggestions. I have added some sentences and explanations to show the uniqueness of the current case.
Minor comments:
6. Abstract - “The bevacizumab regimen is theoretically more effective via two mechanisms: a superior outcome than the standard platinum doublet regimen, confirmed in a phase III trial, and the synergistic effect of bevacizumab with an immune checkpoint inhibitor (ICI).” – I think the meaning of this phrase is difficult to comprehend, especially since the authors state that bevacizumab is more effective, but not compared with what. Please rephrase.
Response:
I appreciate your advice and reworded the relevant part.
7. Case Report – “The patient had undergone right upper lobectomy (pT2bN2M0, stage IIIA) six years preceding the start of this study.” This is not a study. Please rephrase and try to create a more clear timeline.
Response:
I appreciate your advice, and rephrased the expression with clearer timeline.
8. Discussion – “ICI have changed the treatment” instead of “ICI has changed the treatment”
Response:
I appreciate your advice and reworded the relevant part.

Reviewer 3 Report
This indeed is an interesting report showing promising outcome after recurrence of a multimodality treated NSCLC patient. However, some major points have to be addressed before final acceptance can be recommended from my side.
- Was the supraclavicular lymph node the only recurrence? Should be defined.
- Figure 1 states that the patient received definitive chemoradiotherapy, the text says that patient had lobectomy, please clarify
- Was the patient initially (before surgery) cN2 or was stage IIIA defined first after lobectomy with mediastinal lymphadenectomy? Has the patient received induction therapy before surgery due to N2 disease?
- The role of surgery in NSCLC is not clarified in the present manuscript despite the fact, that the patient also underwent lobectomy as part of his multimodality therapy
Line 10&27: “Chemoimmunotherapy has become the standard of care as the first-line treatment of non-small cell lung cancer (NSCLC).” True, but not for all stages of NSCLC, in early stage NSCLC resection is standard of care e.g. Please provide more precise statement here, like standard of care for non-resectable / advanced disease
Line 31: CP should be explained. In general, there are too many abbreviations making the text hardly readable
Line 51: what is S-1?
Line 78: again, ICI is not standard first line treatment for NSCLC with PD-L1 expression >50% over all stages, this is not true for resectable disease.
Author Response
I really appreciate your sincere suggestions and advices.
I have prepared the responses to your concerns in a point-by-point manner, which are attached below.
I am willing to respond as soon as possible if you have further concerns.
This indeed is an interesting report showing promising outcome after recurrence of a multimodality treated NSCLC patient. However, some major points have to be addressed before final acceptance can be recommended from my side.
- Was the supraclavicular lymph node the only recurrence? Should be defined.
Response:
I really appreciate your sincere suggestion. The supraclavicular lymph node was definitely the only recurrence, and I added “locoregional recurrence” to define that this is the only recurrence site.
- Figure 1 states that the patient received definitive chemoradiotherapy, the text says that patient had lobectomy, please clarify
Response:
I really appreciate your advice. I am sorry for the unclear timeline, since the timeline of the current case is complicated. The patient had undergone right upper lobectomy. The recurrence was documented in the right supraclavicular lymph node three years after surgery. Since this was recognized as loco-regional recurrence, the patient underwent definitive concurrent chemoradiotherapy with two cycles of carboplatin plus S-1 (tegafur / gimeracil / oteracil potassium).
- Was the patient initially (before surgery) cN2 or was stage IIIA defined first after lobectomy with mediastinal lymphadenectomy? Has the patient received induction therapy before surgery due to N2 disease?
Response:
I appreciate your suggestion. The patient underwent lobectomy under the diagnosis with cN1, and pN2 was demonstrated after surgery. Then, the patient received adjuvant chemotherapy. The patient did not receive induction therapy before surgery. I added some changes in the timeline of the Case Report section.
- The role of surgery in NSCLC is not clarified in the present manuscript despite the fact, that the patient also underwent lobectomy as part of his multimodality therapy
Response:
I am sorry for the complicated timeline of the case report. As mentioned above, the patient initially underwent lobectomy as a curative surgery, and the locoregional recurrence was observed. I added some changes in the timeline of the Case Report section.
Line 10&27: “Chemoimmunotherapy has become the standard of care as the first-line treatment of non-small cell lung cancer (NSCLC).” True, but not for all stages of NSCLC, in early stage NSCLC resection is standard of care e.g. Please provide more precise statement here, like standard of care for non-resectable / advanced disease
Response:
I thank you for appropriate advice. I added “advanced or recurrent” non-small cell lung cancer.
Line 31: CP should be explained. In general, there are too many abbreviations making the text hardly readable
Response:
I am sorry for too many abbreviations. CP stands for “carboplatin plus paclitaxel”.
It would be better for the readers to explain CP. However, bevacizumab plus CP is abbreviated as “BCP”. I think explaining these abbreviations separately would be redundant, and I adopted this expression.
Line 51: what is S-1?
Response:
I am sorry for incomplete explanation. S-1 stands for “tegafur / gimeracil / oteracil potassium”.
I added the name in full spelling.
Line 78: again, ICI is not standard first line treatment for NSCLC with PD-L1 expression >50% over all stages, this is not true for resectable disease.
Response:
I thank you for appropriate advice. I added “advanced or recurrent” non-small cell lung cancer.
Round 2
Reviewer 3 Report
The authors have addressed all my issues and the manuscript is fine for me.
Best and congratulations